# Chenowethite, Mg(H₂O)₆[(UO₂)₂(SO₄)₂(OH)₂]·5H₂O, a New Mineral with Uranyl-Sulfate Sheets from Red Canyon, Utah, USA

Anthony R. Kampf [1,*], Jakub Plášil [2], Travis A. Olds [3], Chi Ma [4] and Joe Marty [1]

[1] Mineral Sciences Department, Natural History Museum of Los Angeles County, 900 Exposition Boulevard, Los Angeles, CA 90007, USA

[2] Institute of Physics ASCR, v.v.i., Na Slovance 1999/2, 18221 Prague 8, Czech Republic

[3] Section of Minerals and Earth Sciences, Carnegie Museum of Natural History, 4400 Forbes Avenue, Pittsburgh, PA 15213, USA

[4] Division of Geological and Planetary Sciences, California Institute of Technology, 1200 East California Boulevard, Pasadena, CA 91125, USA

\* Correspondence: akampf@nhm.org

**Abstract:** The new mineral chenowethite, Mg(H₂O)₆[(UO₂)₂(SO₄)₂(OH)₂]·5H₂O, was found in efflorescence crusts on tunnel walls at the Blue Lizard, Green Lizard and Markey uranium mines in Red Canyon, San Juan County, Utah, USA. The crystals are long, thin blades up to about 0.5 mm long, occurring in irregular sprays and subparallel groups. Chenowethite is pale green yellow. It has white streak, vitreous to silky luster, brittle tenacity, splintery and stepped fracture and two cleavages: {010} perfect and {001} good. It has a hardness (Mohs) of about 2 and is nonfluorescent in both long- and short-wave ultraviolet illumination. The density is 3.05(2) g/cm³. Optically, crystals are biaxial (−) with α = 1.530(2), β = 1.553(2) and γ = 1.565(2) (white light). The 2V is 72(2)° and dispersion is $r > v$ (slight). The optical orientation is $X$ = **b**, $Y$ = **a**, $Z$ = **c** and the mineral exhibits weak pleochroism in shades of pale green yellow: $X < Y < Z$. The Raman spectrum is consistent with the presence of UO₂²⁺, SO₄²⁻ and OH⁻/H₂O. The empirical formula from electron microprobe analysis and arranged in accordance with the structure is (Mg₀.₇₁Fe²⁺₀.₀₉Co₀.₀₅Ni₀.₀₄)_Σ₀.₈₉(H₂O)₆[(UO₂)₂(SO₄)₂(OH)₂]·[(H₂O)₄.₇₈(NH₄)₀.₂₂]_Σ₅.₀₀. Chenowethite is orthorhombic, space group *Cmcm*; the unit-cell parameters are $a$ = 6.951(2), $b$ = 19.053(6), $c$ = 16.372(5) Å, $V$ = 2168.19(7) Å³ and $Z$ = 4. The crystal structure of chenowethite ($R_1$ = 0.0396 for 912 $I > 2σI$ reflections) contains [(UO₂)₂(SO₄)₂(OH)₂]²⁻ sheets that are topologically equivalent to those in deliensite, feynmanite, greenlizardite, johannite, meitnerite and plášilite.

**Keywords:** chenowethite; new mineral; Raman spectroscopy; crystal structure; uranyl-sulfate sheets; Red Canyon; Utah; USA

## 1. Introduction

Inactive uranium mines have proven prolific sources of new low-temperature uranyl minerals. These phases, typically found in efflorescent coatings on tunnel walls, form from aqueous solutions that permeate and alter the uranium-rich wallrocks. As these solutions, now laden with uranyl complexes and a variety of other cations and anions, seep from the tunnel walls and evaporate, a diverse array of minerals crystallizes. In the past decade, no area in the world has yielded more new low-temperature uranyl mineral species than Red Canyon in southeastern Utah (USA). Until the recent closures of the mines in Red Canyon to collecting, a team led by one of the authors (JM) methodically sampled the secondary mineralization in the Blue Lizard, Green Lizard, Markey and Giveaway-Simplot mines, resulting in the discovery of 38 new uranyl minerals, with several others yet to be described [e.g., [1,2]]. In addition, four new non-uranyl minerals, all sulfates, have also been described. Herein, we describe chenowethite, one of the most recent new uranyl sulfate minerals to be discovered in the Red Canyon mineral assemblages.

Chenowethite is named for American uranium geologist William L. Chenoweth (1928–2018). Dr. Chenoweth received his Ph.D. in geology from the University of New Mexico in 1953. He spent his entire career studying the uranium deposits of the western U.S. He was employed by the U.S. Atomic Energy Commission (AEC), working out of their office in Grants, Arizona until 1964 and then out of their Grand Junction, Colorado office. He was Chief of the Geology Branch in Grand Junction from 1970 until his retirement in 1983, during which time he was responsible for the activities of AEC geologists in the 14 western U.S. states. He then became a uranium mining consultant and a research associate at the New Mexico Bureau of Geology and Mineral Resources. He served as the chairman of the Nuclear Minerals Committee for the Energy Minerals Division of the American Association of Petroleum Geologists from 1983–1998. Dr. Chenoweth authored or co-authored more than 150 reports on the uranium mining history, geology and resources in New Mexico, Arizona, Colorado and Utah. The bulk of the content of these reports stem from his more than 30 years researching uranium for the AEC and the U.S. Department of Energy (DOE). Among these reports is the definitive report on the geology and production history of the uranium deposits of the White Canyon district, which includes the mines in which chenowethite was discovered. At the time of his death, Dr. Chenoweth was still serving as the secretary/treasurer of the Grand Junction Geological Society.

The new mineral and its name were approved by the International Mineralogical Association (IMA) Commission on New Minerals, Nomenclature and Classification (CNMNC) with the number 2022–063. The description is based on four cotype specimens deposited in the collections of the Natural History Museum of Los Angeles County, 900 Exposition Boulevard, Los Angeles, CA 90007, USA, under catalogue numbers 76259 (Blue Lizard mine), 76260 (Blue Lizard mine), 76261 (Green Lizard mine) and 76262 (Green Lizard mine).

## 2. Occurrence, Geological Setting and Mineral Association

Chenowethite was discovered on specimens collected underground in the Blue Lizard mine (37°33′26″ N 110°17′44″ W), the Green Lizard mine (37°34′37.10″ N 110°17′52.80″ W) and the Markey mine (37°32′57″ N 110°18′08″ W) in the White Canyon mining district, San Juan County, Utah, USA. The foregoing description is based only on specimens from the Blue Lizard and Green Lizard mines and only these should be considered type localities for the species. The Blue Lizard mine is on the west-facing side of Red Canyon and the Markey mine is on the east-facing side about 1 km southwest of the Blue Lizard mine. The Green Lizard mine is near the head of Low Canyon on the east side of Red Canyon, 2.1 km north of the Blue Lizard mine. The geology of all three mines is very similar [3,4], although the secondary mineralogy of the Green Lizard mine is richer in ammonium phases and that of the Markey mine is richer in carbonate phases. The following information on the mines and their geology is taken largely from [3].

Mineralized channels are in the Shinarump member of the Chinle Formation. The Shinarump member consists of medium- to coarse-grained sandstone, conglomeratic sandstone beds and thick siltstone lenses. Ore minerals were deposited as replacements of wood and other organic material and as disseminations in the enclosing sandstone. Since the mine closed, oxidation of primary ores in the humid underground environment has produced a variety of secondary minerals, mainly sulfates, as efflorescent crusts on the surfaces of mine walls.

Chenowethite is a relatively rare mineral found in association with ammoniozippeite, dickite, gypsum, hexahydrite, johannite and plášilite at the Blue Lizard mine. At the Green Lizard mine, it is found in association with dickite, gypsum, natrojarosite, natrozippeite and johannite.

Uranyl sulfate minerals typically form by hydration–oxidation weathering of primary uranium minerals, mainly uraninite, by acidic solutions derived from the decomposition of associated sulfides [5–7]. Chenowethite and other secondary minerals occurring in the efflorescent crusts of the mines of Red Canyon were formed by such a process.

### 3. General Appearance, Physical, Chemical and Optical Properties

Chenowethite occurs as long, thin blades up to about 0.5 mm long, forming irregular sprays and subparallel groups (Figure 1). The blades are flattened on {010} and elongated parallel to [100]. The observed crystal forms are {010}, {001} and {101} (Figure 2). No twinning was observed. The color of the mineral is pale green yellow. It has white streak, vitreous to silky luster, brittle tenacity, splintery and stepped fracture and two cleavages: {010} perfect and {001} good. The Mohs hardness based on scratch tests is about 2. The mineral is nonfluorescent in both long- and short-wave ultraviolet illumination. The density measured via flotation in a mixture of methylene iodide and toluene is 3.05(2) g/cm$^3$. The calculated density based on the empirical formula and single-crystal cell is 3.045 g/cm$^3$. The mineral is easily soluble in room-temperature $H_2O$.

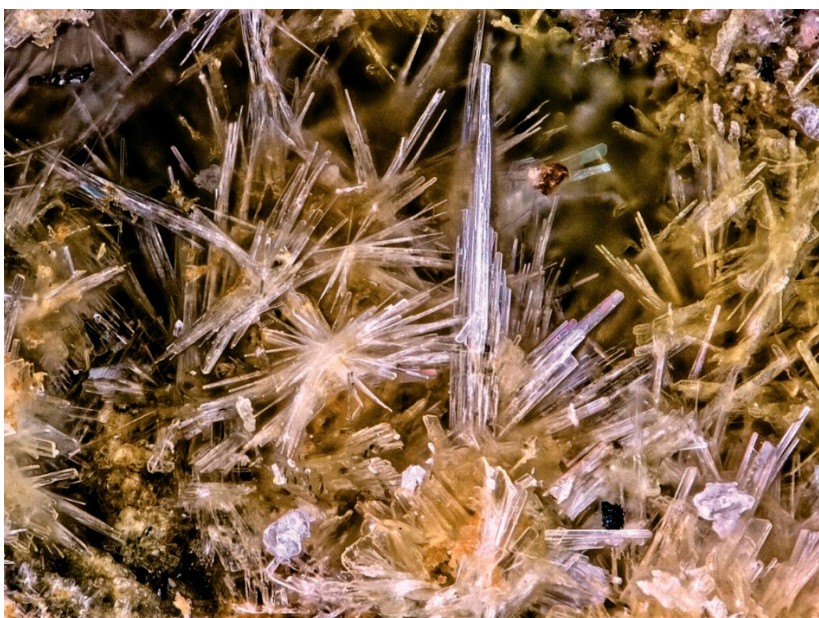

**Figure 1.** Chenowethite blades on cotype specimen #76260 from the Blue Lizard mine. The field of view is 0.84 mm across.

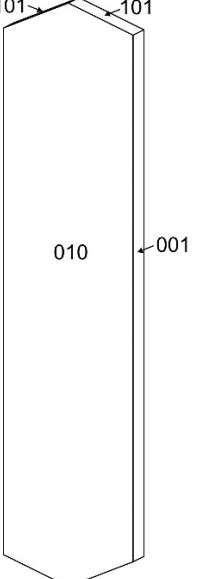

**Figure 2.** Crystal drawing of chenowethite; clinographic projection in non-standard orientation, [100] vertical.

Optically, crystals are biaxial ($-$) with $\alpha = 1.530(2)$, $\beta = 1.553(2)$ and $\gamma = 1.565(2)$, measured in white light. The $2V$ measured directly by conoscopic observation of the interference figure on a spindle stage is $72(2)°$, The $2V$ calculated from the indices of refraction is $70.7°$. The dispersion was observed to be $r > v$, slight. The optical orientation is $X = \mathbf{b}$, $Y = \mathbf{a}$, $Z = \mathbf{c}$. The mineral exhibits weak pleochroism in shades of pale green yellow: $X < Y < Z$.

The Gladstone–Dale compatibility [8] $1 - (K_p/K_c)$ is 0.004 (superior) based on the empirical formula and $-0.002$ (superior) based on the ideal formula, in both cases using $k(UO_3) = 0.118$ [9].

## 4. Raman Spectroscopy

Raman spectroscopy was performed via a Horiba XploRa Plus micro-Raman spectrometer (Kyoto, Japan) using an incident wavelength of 532 nm, laser slit of 100 μm, 1800 gr/mm diffraction grating and a $100\times$ (0.9 NA) objective. The spectrum, recorded from 4000 to 100 cm$^{-1}$, is shown in Figure 3.

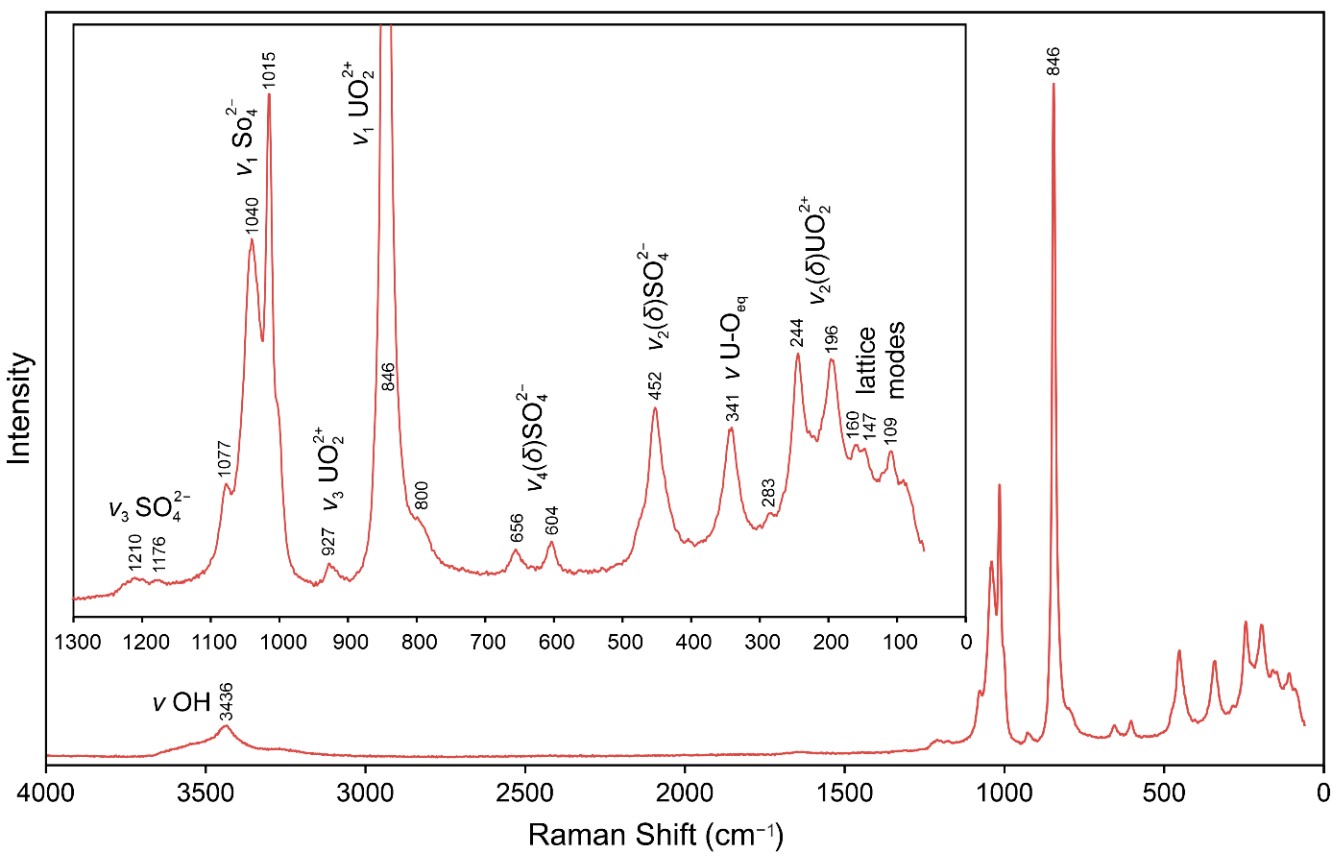

**Figure 3.** Raman spectrum of chenowethite recorded using a 532 nm diode laser.

The band at 3436 cm$^{-1}$ with broad shoulders extending from about 3700 to 3200 cm$^{-1}$ is due to ν OH stretching vibrations. Using the empirically derived equation of [10], the calculated O$\cdots$O distance for the hydrogen bond corresponding to the 3436 cm$^{-1}$ band is 2.82 Å. The complex nature of the interlayer portion of the structure, which includes several different $H_2O$ groups, some of which are disordered, clearly involves numerous hydrogen bonds covering a wide range of O$\cdots$O distances. This accounts for the broad shoulders in the ν OH region of the spectrum and it makes the unambiguous assignment of the specific hydrogen bond corresponding to the 3436 cm$^{-1}$ band impossible. A barely perceptible band at about 1630–1650 cm$^{-1}$ is presumably related to the $\nu_2(\delta)$ bending vibrations of $H_2O$.

The numerous bands in the 1300 to 190 cm$^{-1}$ portion of the spectrum are assignable to a variety of vibrational modes of $SO_4^{2-}$ and $UO_2^{2+}$ as labeled in Figure 3. According to the empirical relationship of [11], the $\nu_1$ $UO_2^{2+}$ and $\nu_3$ $UO_2^{2+}$ vibrations at 846 and 927 cm$^{-1}$, respectively, both correspond to an approximate U–O$_{Ur}$ bond length of 1.77 Å, in excellent agreement with U–O$_{Ur}$ bond lengths from the X-ray data: 1.756(12) and 1.766(12) Å.

## 5. Chemical Composition

Analyses (eight points) were performed at Caltech on a JEOL 8200 electron microprobe in WDS mode. Analytical conditions were 15 kV accelerating voltage, 10 nA beam current and a beam diameter of 5 µm. The blades of chenowethite are very fragile and impossible to polish; consequently, analyses were performed on unpolished crystal faces. Because insufficient material is available for a direct determination of $H_2O$, it was calculated based upon the structure determination (S = 2 and O + N = 25 *apfu*). The high analytical total is attributed to water loss under vacuum. The analytical results are given in Table 1.

**Table 1.** Chemical composition (wt%) for chenowethite based on one analysis.

| Constituent | Mean | Range | S.D. | Standard |
|---|---|---|---|---|
| $(NH_4)_2O$ | 0.60 | 0.43–0.70 | 0.09 | BN |
| MgO | 2.96 | 2.50–3.46 | 0.35 | forsterite |
| FeO | 0.67 | 0.52–0.94 | 0.15 | fayalite |
| CoO | 0.35 | 0.30–0.46 | 0.05 | Co metal |
| NiO | 0.29 | 0.24–0.34 | 0.04 | NiO |
| $SO_3$ | 16.61 | 15.73–17.33 | 0.59 | anhydrite |
| $UO_3$ | 59.33 | 58.42–60.07 | 0.58 | $UO_2$ |
| $H_2O$ * | 22.03 | | | |
| Total | 102.84 | | | |

*: based on structure.

The empirical formula (based on 2 S *apfu*) is $(NH_4)_{0.22}Mg_{0.71}Fe^{2+}{}_{0.09}Co_{0.05}Ni_{0.04}U_{2.00}S_2$ $O_{24.78}H_{23.58}$ or, arranged structurally, $(Mg_{0.71}Fe^{2+}{}_{0.09}Co_{0.05}Ni_{0.04})_{\sum 0.89}(H_2O)_6[(UO_2)_2(SO_4)_2$ $(OH)_2]\cdot[(H_2O)_{4.78}(NH_4)_{0.22}]_{\sum 5.00}$. The simplified formula is $(Mg,Fe,Co,Ni,\square)(H_2O)_6[(UO_2)_2$ $(SO_4)_2(OH)_2]\cdot5(H_2O,NH_4)$; or $[(Mg,Fe,Co,Ni)^{2+}{}_{1-x}\square_x](H_2O)_6[(UO_2)_2(SO_4)_2(OH)_2]\cdot5[(H_2O)_{1-2x}$ $(NH_4)^+{}_{2x}]$. The ideal formula is $Mg(H_2O)_6[(UO_2)_2(SO_4)_2(OH)_2]\cdot5H_2O$, which requires MgO 4.08, $UO_3$ 57.86, $SO_3$ 16.20, $H_2O$ 21.87, total 100 wt.%. As explained below, the $H_2O$ following the dot in the formula corresponds to isolated $H_2O$ groups and the small amount of N is presumed to be present as $NH_4^+$ substituting for one or more of the isolated $H_2O$ groups. The $NH_4^+$ serves to compensate for the charge deficiency resulting from the partial vacancy in the Mg site.

## 6. X-ray Diffraction

X-ray powder diffraction data were recorded using a Rigaku R-Axis Rapid II curved imaging plate microdiffractometer (Tokyo, Japan) with monochromatized Mo*K*α radiation. A Gandolfi-like motion on the φ and ω axes was used to randomize the sample. Observed *d* values and intensities were derived by profile fitting using JADE Pro software (Materials Data Inc., Livermore, CA, USA). Data are given in Table 2. The orthorhombic (space group *Cmcm*) unit-cell parameters refined from the powder data using JADE Pro with whole pattern fitting are *a* = 6.951(2), *b* = 19.053(6), *c* = 16.372(5) Å and *V* = 2168.19(7) Å$^3$.

**Table 2.** Powder X-ray data (*d* in Å) for chenowethite. Only calculated lines with $I \geq 1.5$ are listed.

| $I_{obs}$ | $d_{obs}$ | $d_{calc}$ | $I_{calc}$ | *hkl* | $I_{obs}$ | $d_{obs}$ | $d_{calc}$ | $I_{calc}$ | *hkl* |
|---|---|---|---|---|---|---|---|---|---|
| 100 | 9.54 | 9.5175 | 100 | 0 2 0 | 13 | 2.278 | 2.2775 | 2 | 0 6 5 |
| 15 | 8.21 | 8.2264 | 11 | 0 2 1 | | | 2.2752 | 4 | 3 1 1 |
| | | 8.1789 | 2 | 0 0 2 | | | 2.1999 | 2 | 1 1 7 |
| 80 | 6.07 | 6.0591 | 41 | 1 1 1 | 17 | 2.175 | 2.1808 | 2 | 0 8 3 |
| | | 4.7588 | 16 | 0 4 0 | | | 2.1744 | 3 | 3 3 0 |
| 54 | 4.712 | 4.7312 | 3 | 0 2 3 | | | 2.1554 | 2 | 3 3 1 |
| | | 4.6839 | 21 | 1 3 0 | 21 | 2.141 | 2.1528 | 8 | 1 7 4 |
| 34 | 4.535 | 4.5693 | 13 | 0 4 1 | | | 2.1293 | 5 | 2 4 5 |
| | | 4.5029 | 13 | 1 3 1 | 14 | 2.110 | 2.1173 | 4 | 3 1 3 |
| 15 | 4.183 | 4.1835 | 7 | 1 1 3 | | | 2.0910 | 2 | 1 3 7 |
| 16 | 4.094 | 4.0894 | 13 | 0 0 4 | 13 | 2.033 | 2.0447 | 4 | 0 0 8 |
| 29 | 3.762 | 3.7573 | 22 | 0 2 4 | | | 2.0232 | 3 | 1 9 0 |
| | | 3.5853 | 4 | 0 4 3 | 10 | 1.9894 | 1.9777 | 3 | 3 5 0 |
| | | 3.5530 | 2 | 1 3 3 | 6 | 1.9378 | 1.9487 | 3 | 2 8 1 |
| 41 | 3.476 | 3.4718 | 15 | 2 0 0 | | | 1.9243 | 2 | 0 8 5 |
| | | 3.4648 | 4 | 1 1 4 | | | 1.9198 | 3 | 3 3 4 |
| 14 | 3.336 | 3.3382 | 11 | 1 5 0 | 15 | 1.9081 | 1.9043 | 5 | 2 6 5 |
| 28 | 3.259 | 3.2615 | 15 | 2 2 0 | | | 1.8907 | 2 | 0 10 1 |
| | | 3.1986 | 3 | 2 2 1 | 20 | 1.8798 | 1.8802 | 5 | 3 1 5 |
| | | 3.1145 | 5 | 0 6 1 | | | 1.8739 | 3 | 1 3 8 |
| 25 | 3.102 | 3.1015 | 7 | 0 4 4 | 10 | 1.8135 | 1.8134 | 4 | 1 9 4 |
| | | 3.0939 | 2 | 0 2 5 | | | 1.8109 | 2 | 3 3 5 |
| | | 3.0805 | 6 | 1 3 4 | 10 | 1.7792 | 1.7804 | 4 | 3 5 4 |
| 27 | 2.928 | 2.9244 | 19 | 1 1 5 | 18 | 1.7565 | 1.7625 | 7 | 3 7 0 |
| 10 | 2.799 | 2.8047 | 7 | 2 4 0 | | | 1.7508 | 6 | 1 1 9 |
| | | 2.7643 | 2 | 2 4 1 | | | 1.7436 | 3 | 1 5 8 |
| | | 2.7421 | 2 | 0 6 3 | 17 | 1.7364 | 1.7359 | 2 | 4 0 0 |
| 9 | 2.684 | 2.6821 | 7 | 1 3 5 | | | 1.7324 | 3 | 2 2 8 |
| 32 | 2.650 | 2.6466 | 15 | 2 0 4 | 14 | 1.6999 | 1.7077 | 2 | 4 2 0 |
| 21 | 2.560 | 2.5860 | 6 | 1 5 4 | | | 1.6944 | 3 | 1 3 9 |
| | | 2.5499 | 12 | 2 2 4 | 9 | 1.6750 | 1.6830 | 2 | 2 8 5 |
| 15 | 2.534 | 2.5320 | 8 | 1 7 0 | 9 | 1.6196 | 1.6186 | 3 | 3 7 4 |
| | | 2.4941 | 2 | 2 4 3 | 17 | 1.5913 | 1.5979 | 2 | 4 0 4 |
| 7 | 2.352 | 2.3562 | 2 | 1 3 6 | | | 1.5908 | 3 | 1 7 8 |
| | | 2.3546 | 3 | 0 8 1 | 8 | 1.5671 | 1.5758 | 3 | 4 2 4 |
| 11 | 2.315 | 2.3183 | 2 | 2 6 1 | | | | | |
| | | 2.3130 | 4 | 2 4 4 | | | | | |

Single-crystal X-ray studies were performed using the same diffractometer and radiation noted above. The Rigaku CrystalClear software package (Tokyo, Japan) was used for processing the structure data, including the application of an empirical absorption correction using the multi-scan method with ABSCOR [12]. The structure was solved using the intrinsic phasing algorithm of SHELXT [13]. SHELXL-2016 [14] was used for the refinement of the structure. The octahedrally coordinated Mg site was initially located on the special position 0,0,0; however, anisotropic refinement of the site indicated it to be split along with its coordinated O sites (OW1 and OW2) (Figure 4). The EMPA-determined cation content $Mg_{0.71}Fe^{2+}_{0.09}Co_{0.05}Ni_{0.04}\square_{0.11}$ was assigned to the split Mg site. Although the site-scattering value calculated for this occupancy (53.3) was significantly lower than the refined site-scattering value for the site (61.9), the resultant $R_1$ was almost identical for the two refinements (0.0396 vs. 0.0395). Difference Fourier syntheses failed to locate possible H atom positions. Data collection and refinement details are given in Table 3, atom coordinates and displacement parameters in Table 4, selected bond distances in Table 5 and a bond valence analysis [15,16] in Table 6.

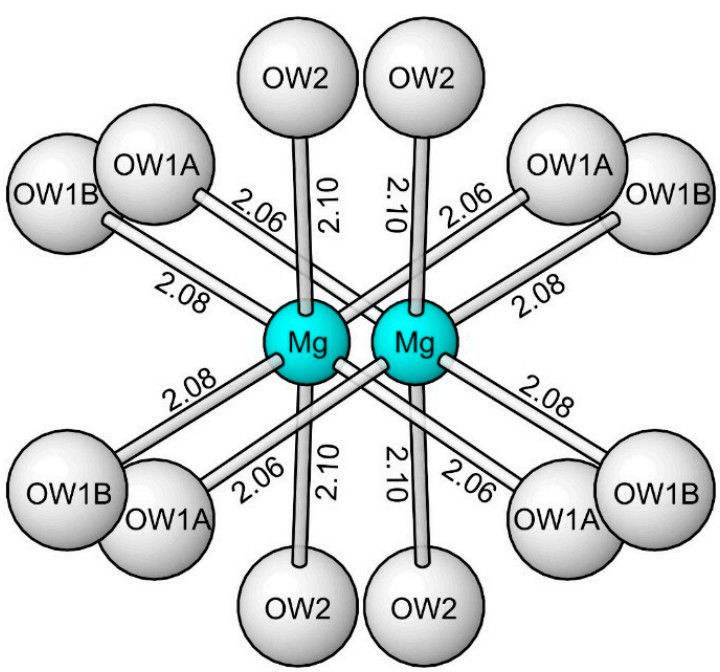

**Figure 4.** The coordination of the split Mg site in the structure of chenowethite.

**Table 3.** Data collection and structure refinement details for chenowethite.

| | |
|---|---|
| Diffractometer | Rigaku R-Axis Rapid II |
| X-ray radiation/power | Mo$K\alpha$ ($\lambda$ = 0.71075 Å)/50 kV, 40 mA |
| Temperature | 293(2) K |
| Structural formula | $(Mg_{0.71}Fe^{2+}{}_{0.09}Co_{0.05}Ni_{0.04})_{\sum 0.89}U_2S_2O_{25}$ |
| Space group | *Cmcm* (#63) |
| Unit cell dimensions | *a* = 6.9435(6) Å |
| | *b* = 19.035(2) Å |
| | *c* = 16.3577(13) Å |
| *V* | 2162.0(3) Å$^3$ |
| *Z* | 4 |
| Density (for above formula) | 2.974 g cm$^{-3}$ |
| Absorption coefficient | 15.414 mm$^{-1}$ |
| *F*(000) | 1718 |
| Crystal size | 100 × 50 × 7 μm |
| θ range | 3.12 to 25.01° |
| Index ranges | $-8 \leq h \leq 8, -22 \leq k \leq 22, -18 \leq l \leq 19$ |
| Reflections collected/unique | 5708/1071; $R_{\text{int}}$ = 0.083 |
| Reflections with $I > 2\sigma I$ | 912 |
| Completeness to θ = 25.01° | 99.5% |
| Refinement method | Full-matrix least-squares on $F^2$ |
| Parameter/restraints | 103/0 |
| GoF | 1.080 |
| Final *R* indices [$F > 4\sigma(F)$] | $R_1$ = 0.0396, $wR_2$ = 0.0929 |
| *R* indices (all data) | $R_1$ = 0.0482, $wR_2$ = 0.0982 |
| Largest diff. peak/hole | +2.69/−1.05 e A$^{-3}$ |

$R_{\text{int}} = \Sigma |F_o{}^2 - F_o{}^2(\text{mean})| / \Sigma [F_o{}^2]$. GoF = $S = \{\Sigma[w(F_o{}^2 - F_c{}^2)^2]/(n - p)\}^{1/2}$. $R_1 = \Sigma ||F_o| - |F_c||/\Sigma|F_o|$. $wR_2 = \{\Sigma[w(F_o{}^2 - F_c{}^2)^2]/\Sigma[w(F_o{}^2)^2]\}^{1/2}$; $w = 1/[\sigma^2(F_o{}^2) + (aP)^2 + bP]$ where $a$ is 0.049, $b$ is 23.3969 and $P$ is $[2F_c{}^2 + \text{Max}(F_o{}^2,0)]/3$.

**Table 4.** Atom coordinates and displacement parameters ($\text{Å}^2$) for chenowethite.

| | *x/a* | *y/b* | *z/c* | $U_{eq}$ | Occupancy | |
|---|---|---|---|---|---|---|
| Mg | 0.9545(10) | 0 | 0 | 0.025(3) | $Mg_{0.71}Fe^{2+}_{0.09}Co_{0.05}Ni_{0.04}$ | |
| U | 0.5 | 0.21326(3) | 0.13177(3) | 0.0173(2) | 1 | |
| S | 0 | 0.2545(2) | 0.08766(19) | 0.0187(7) | 1 | |
| O1 | 0 | 0.2972(5) | 0.1605(6) | 0.028(2) | 1 | |
| O2 | 0 | 0.3002(5) | 0.0144(5) | 0.025(2) | 1 | |
| O3 | 0.1704(8) | 0.2075(3) | 0.0840(4) | 0.0231(15) | 1 | |
| O4 | 0.5 | 0.3052(6) | 0.1237(5) | 0.028(2) | 1 | |
| O5 | 0.5 | 0.1207(6) | 0.1378(4) | 0.024(3) | 1 | |
| OH | 0.3078(14) | 0.2190(5) | 0.25 | 0.024(2) | 1 | |
| OW1A | 0.174(4) | 0.0727(16) | 0.9986(17) | 0.049(7) | 0.5 | |
| OW1B | 0.247(4) | 0.0778(18) | 0.9757(18) | 0.056(7) | 0.5 | |
| OW2 | 0.947(4) | 0.0127(9) | 0.1274(7) | 0.059(11) | 0.5 | |
| OW3 | 0 | 0.1136(8) | 0.25 | 0.036(4) | 1 | |
| OW4 | 0.274(3) | 0.4095(9) | 0.25 | 0.097(6) | 1 | |
| OW5 | 0.566(3) | 0.9652(13) | 0.1240(10) | 0.075(7) | 0.5 | |
| | $u^{11}$ | $u^{22}$ | $u^{33}$ | $u^{23}$ | $u^{13}$ | $u^{12}$ |
| Mg | 0.023(8) | 0.024(4) | 0.027(3) | −0.009(3) | 0 | 0 |
| U | 0.0127(3) | 0.0295(4) | 0.0099(3) | 0.0015(2) | 0 | 0 |
| S | 0.0119(15) | 0.033(2) | 0.0108(16) | 0.0016(14) | 0 | 0 |
| O1 | 0.029(6) | 0.040(7) | 0.015(5) | −0.007(4) | 0 | 0 |
| O2 | 0.015(5) | 0.044(7) | 0.017(5) | 0.006(4) | 0 | 0 |
| O3 | 0.013(3) | 0.037(4) | 0.019(3) | −0.005(3) | −0.002(3) | 0.001(3) |
| O4 | 0.014(5) | 0.052(8) | 0.018(5) | −0.003(4) | 0 | 0 |
| O5 | 0.003(4) | 0.063(8) | 0.006(4) | −0.004(4) | 0 | 0 |
| OH | 0.022(5) | 0.046(7) | 0.005(4) | 0 | 0 | 0.002(4) |
| OW1A | 0.07(2) | 0.037(13) | 0.035(14) | 0.000(11) | 0.014(12) | −0.028(13) |
| OW1B | 0.07(2) | 0.043(13) | 0.050(18) | 0.003(12) | 0.012(12) | −0.030(15) |
| OW2 | 0.10(3) | 0.047(10) | 0.027(7) | −0.004(6) | −0.013(8) | −0.011(11) |
| OW3 | 0.048(9) | 0.031(9) | 0.029(8) | 0 | 0 | 0 |
| OW4 | 0.111(14) | 0.069(13) | 0.111(14) | 0 | 0 | −0.003(10) |
| OW5 | 0.08(2) | 0.073(15) | 0.070(12) | 0.005(9) | 0.011(9) | −0.001(11) |

**Table 5.** Selected bond distances ($\text{Å}$) for chenowethite.

| | | | | Hydrogen bonds * | |
|---|---|---|---|---|---|
| U–O4 | 1.756(12) | S–O1 | 1.442(11) | | |
| U–O5 | 1.765(12) | S–O2 | 1.482(10) | OH⋯O1 | 2.987(11) |
| U–OH (×2) | 2.352(5) | S–O3 (×2) | 1.484(6) | OW1A⋯O3 | 2.92(3) |
| U–O2 | 2.405(9) | <S–O> | 1.473 | OW1B⋯O2 | 2.92(3) |
| U–O3 (×2) | 2.421(6) | | | OW3⋯OH | 2.931(15) |
| <U–O$_{Ur}$> | 1.761 | Mg–OW1A (×2) | 2.06(3) | OW4⋯O4 | 3.265(17) |
| <U–O$_{eq}$> | 2.390 | Mg–OW1B (×2) | 2.08(3) | OW5⋯O5 | 3.00(3) |
| | | Mg–OW2 (×2) | 2.099(12) | | |
| | | <Mg–O> | 2.08 | | |

*: only likely hydrogen bonds to O atoms in the uranyl-sulfate sheet are included.

**Table 6.** Bond-valence analysis for chenowethite. Values are expressed in valence units *.

| | Mg | U | S | Hydrogen Bonds | | Σ |
| | | | | Accepted | Donated | |
|---|---|---|---|---|---|---|
| O1 | | | 1.62 | 0.13$^{\times 2 \rightarrow}$ | | 1.88 |
| O2 | | 0.47 | 1.46 | 0.07 | | 2.00 |
| O3 | | 0.45$^{\times 2\downarrow}$ | 1.46$^{\times 2\downarrow}$ | 0.07 | | 1.98 |
| O4 | | 1.85 | | 0.10 | | 1.95 |
| O5 | | 1.81 | | 0.13 | | 1.94 |
| OH | | 0.52$^{\times 2\downarrow\rightarrow}$ | | 0.15 | −0.13 | 1.06 |
| OW1A | 0.36$^{\times 2\downarrow}$ | | | | | |
| OW1B | 0.34$^{\times 2\downarrow}$ | | | | | |
| OW2 | 0.33$^{\times 2\downarrow}$ | | | | | |
| Σ | 2.06 | 6.07 | 6.00 | | | |

*: multiplicity is indicated by $^{\times\downarrow\rightarrow}$. Bond-valence parameters are from [15]. Hydrogen bond strengths based on O–O bond lengths from [16]. Hydrogen bond contributions to O atoms in the uranyl-sulfate sheet are included.

## 7. Description of Crystal Structure

The U sites in the structure of chenowethite are surrounded by seven O atoms forming a squat $UO_7$ pentagonal bipyramid. This is the most typical coordination for $U^{6+}$, particularly in uranyl sulfates, where the two short apical bonds of the bipyramid constitute the $UO_2$ uranyl group. In the structure of chenowethite, pairs of pentagonal bipyramids share a common edge, forming dimers. The dimers are linked by sharing corners with $SO_4$ groups, yielding a $[(UO_2)_2(SO_4)_2(OH)_2]^{2-}$ sheet parallel to {010} (Figure 5). This sheet is based on the phosphuranylite anion topology [17], with ring symbol $6^15^24^23^2$ [18]. The sheets in deliensite, $Fe(H_2O)_5[(UO_2)_2(SO_4)_2(OH)_2]\cdot2H_2O$ [19]; feynmanite, $Na_2(H_2O)_7[(UO_2)_2(SO_4)_2(OH)_2]$ [20]; greenlizardite, $(NH_4)Na(H_2O)_3[(UO_2)_2(SO_4)_2(OH)_2]\cdot H_2O$ [21]; johannite $Cu(H_2O)_4[(UO_2)_2(OH)_2(SO_4)_2]\cdot4H_2O$ [22]; meitnerite, $(NH_4)_2(H_2O)_4[(UO_2)_2(SO_4)_2(OH)_2]$ [23]; and plášilite, $Na_2(H_2O)_4[(UO_2)_2(OH)_2(SO_4)_2]$ [4], are topologically identical to those in chenowethite. However, these sheets are geometrical isomers, differing in the orientation of the $SO_4$ groups. The sheets in chenowethite and deliensite are the same geometrical isomer (ddudd/uuduu), differing in relatively minor canting of the tetrahedra.

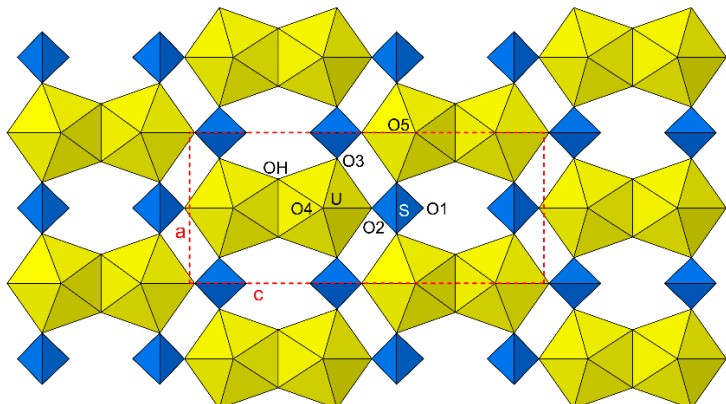

**Figure 5.** The uranyl-sulfate sheet in the structure of chenowethite viewed down [010]. The unit cell outline is shown by dashed red lines with cell directions labeled in red.

The interlayer region in the chenowethite structure contains a disordered $Mg(H_2O)_6$ octahedron and three isolated $H_2O$ groups (OW3, OW4 and OW5). The OW5 site is another split site. The small amount of N noted in the EPMA is presumed to represent $NH_4^+$, which is accommodated in one or more of the isolated $H_2O$ sites. This $NH_4^+$ serves to compensate the charge deficiency resulting from the partial vacancy in the Mg site noted above. Note that $NH_4^+$ is a relatively common constituent in the secondary phases of the

uranium mines in Red Canyon and is an essential constituent in ammoniozippeite, which is intimately associated with chenowethite at the Blue Lizard mine.

All of the structures with $[(UO_2)_2(SO_4)_2(OH)_2]^{2-}$ sheets noted contain cations in their interlayer regions. In the structures of feynmanite, greenlizardite, johannite, meitnerite and plášilite, the interlayer cations serve to link $[(UO_2)_2(SO_4)_2(OH)_2]^{2-}$ sheets to each other via cation–oxygen bonds. In the structure of deliensite, the interlayer Fe-centered octahedron shares one O corner with the apical vertex of a $UO_7$ pentagonal bipyramid in only one sheet, such that adjacent sheets are only linked via hydrogen bonds. In the structure of chenowethite, the interlayer $Mg(H_2O)_6$ octahedron only links to the $[(UO_2)_2(SO_4)_2(OH)_2]^{2-}$ sheets via hydrogen bonds. The structures of chenowethite and deliensite are compared in Figure 6.

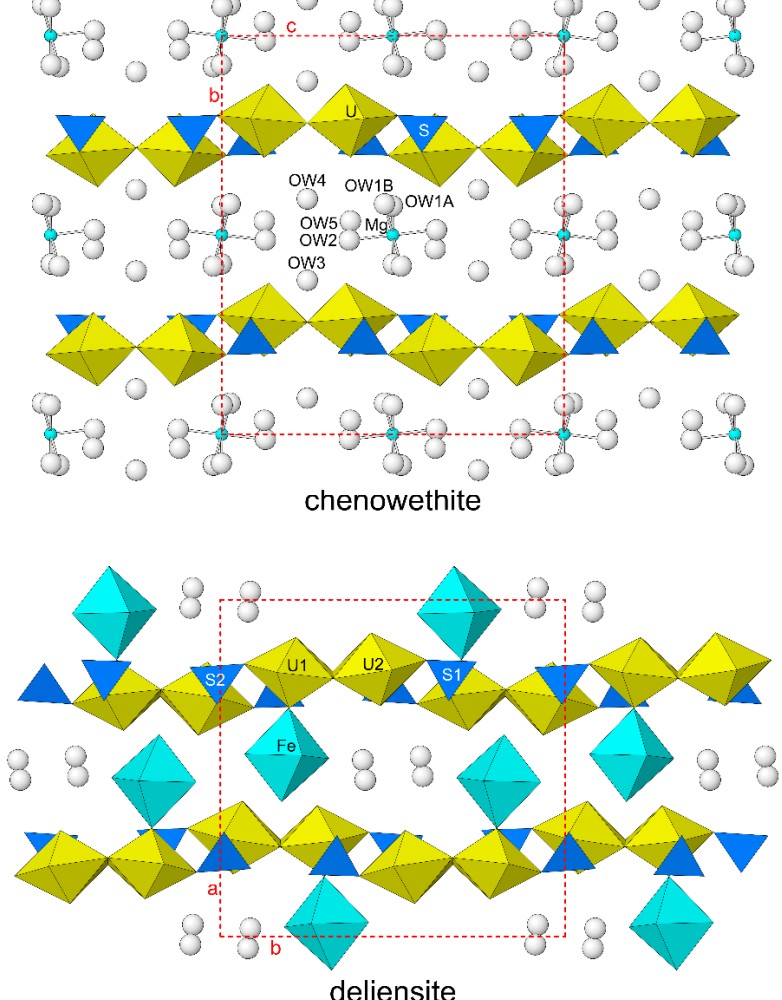

**Figure 6.** The structures of chenowethite and deliensite. The unit cell outlines are shown by dashed red lines with cell directions labeled in red.

## 8. Discussion

Our observations of uranyl sulfates in both the field and laboratory have shown them to occur as complex admixtures of species, often so visually, chemically, or spectroscopically similar to one another that identification is difficult except through a combination of methods. The high structural and chemical diversity of uranyl sulfates, which is attributed to minute variabilities in formation conditions, has greatly advanced our understanding of uranyl-sulfate mineralogy. In general, solution conditions such as ionic strength, pH, and evaporation rate, which depends greatly on local relative humidity, are dominant contributors to the diversity of species in Red Canyon.

The same conditions that drive uranyl-sulfate mineral formation also dictate anthropogenic processes, where, for example, uranyl-sulfate phases crystallized from leaking U-bearing wastes in a geological repository will be important indicators of local solution chemistry and can aid in failure analyses or diagnoses. Thus, continued research on uranyl-sulfate mineral formation in abandoned uranium mines can translate to greater control of uranium in the lab, and may eventually be used to improve industrial techniques for U-waste disposal or remediation of U contaminated sites. Most uranyl-sulfate species described from Red Canyon have yet to be reproduced synthetically; however, the high solubilities of chenowethite and others lend themselves to relatively simple evaporative synthesis techniques. Such studies could produce years of interesting results that supplement our understanding of natural and industrial processes involving uranyl sulfates.

**Author Contributions:** Conceptualization, A.R.K., J.P. and T.A.O.; methodology, A.R.K., J.P., T.A.O., C.M. and J.M.; investigation, A.R.K., J.P., T.A.O., C.M. and J.M.; original manuscript—draft preparation, A.R.K., J.P. and T.A.O.; manuscript—review and editing, A.R.K., J.P., T.A.O., C.M. and J.M.; figures, A.R.K. All authors have read and agreed to the published version of the manuscript.

**Funding:** This study was funded, in part, by the John Jago Trelawney Endowment to the Mineral Sciences Department of the Natural History Museum of Los Angeles County.

**Data Availability Statement:** The crystallographic information file (CIF) for this paper (deposition number 2226329) are provided free of charge by the joint Cambridge Crystallographic Data Centre and Fachinformationszentrum Karlsruhe Access Structures service www.ccdc.cam.ac.uk/structures (accessed on 5 December 2022).

**Conflicts of Interest:** The authors declare no conflict of interest.

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
