# Peer review of "Chenowethite, Mg(H2O)6[(UO2)2(SO4)2(OH)2]·5H2O, a New Mineral with Uranyl-Sulfate Sheets from Red Canyon, Utah, USA"

_minerals, doi:10.3390/min12121594_

Round 1

Reviewer 1 Report

Authors have presented a new mineral found in efflorescence crusts in uranium mines in Red Canyon, Utah, USA. Only a few minor changes should be made:

·         Line 143, page 4: Instead of “According to the empirical relationship of [11]” please explain in your words what you used from this reference, to be easier for a reader to follow.

·       In Figure 5. And Figure 6. The atoms could be noted more clear.

Author Response

Line 143, page 4: Instead of “According to the empirical relationship of [11]” please explain in your words what you used from this reference, to be easier for a reader to follow.
Response: We think that this sentence is perfectly understandable.

In Figure 5. And Figure 6. The atoms could be noted more clear.
Response:  We have increased the font size.

Reviewer 2 Report

This is an interesting report for a new uranyl-sulfate mineral of chenowethite, the authors carried out multiple characterization on it, including Raman, electron microprobe, power and single-crystal XRD. The background of geological occurrence and mineral properties are also well introduced. Overall, this work deserves publication after some revisions. Please refer to my suggestions below:

1) In the formula (like in L24 in the abstract). H2O are divided into two part: (H2O)6 ahead and (H2O)4.78 together with (NH4)0.22 at the end of the formula. Hence, what is the difference between them? It is not clearly exhibited in the text.

2) should nitrogen exist in the form of small NH3 molecule or (NH4)+ cation? It is strange no such signal has been detected in the Raman spectrum.

3) Could the authors tell positions for the group of H2O and (NH4)+ in the crystal structures?

4) If possible, I suggest the authors took an Infrared spectrum (maybe just mid-FTIR spectrum is also helpful), which could give us important information about (OH)- and (NH4)+

Author Response

This is an interesting report for a new uranyl-sulfate mineral of chenowethite, the authors carried out multiple characterization on it, including Raman, electron microprobe, power and single-crystal XRD. The background of geological occurrence and mineral properties are also well introduced. Overall, this work deserves publication after some revisions. Please refer to my suggestions below:

1) In the formula (like in L24 in the abstract). H2O are divided into two part: (H2O)6 ahead and (H2O)4.78 together with (NH4)0.22 at the end of the formula. Hence, what is the difference between them? It is not clearly exhibited in the text.
Response: Two sentences have been added to explain this.

2) should nitrogen exist in the form of small NH3 molecule or (NH4)+ cation? It is strange no such signal has been detected in the Raman spectrum.
Response: A sentence has been added to explain why we assume the N to be present as NH4+. Considering the very small amount of NH4+ present in the mineral, it is not at all surprising that there is no evidence for it in the Raman spectrum.

3) Could the authors tell positions for the group of H2O and (NH4)+ in the crystal structures?
Response: The isolated H2O sites OW3, OW4 and OW5 all appear to be capable of hosting NH4+.

4) If possible, I suggest the authors took an Infrared spectrum (maybe just mid-FTIR spectrum is also helpful), which could give us important information about (OH)- and (NH4)+.
Response: We are confident of our conclusions based on the data provided and we do not think that FTIR would be particularly useful.

Reviewer 3 Report

This is a well-formed manuscript dedicated to the discovery and mineralogical and crystal-chemical description of the magnesian uranyl sulfate hydrate hydroxide, named chenowethite. The article is traditionally formatted and includes a detailed presentation of the morphology and optical properties of the new mineral, as well as data of electron microprobe analysis and X-ray powder diffraction. The determination of the crystal structure from a single crystal is characterized by low uncertainties, which allowed the authors to carry out a comprehensive crystal chemical analysis of chenowethite in comparison with related minerals of a similar composition.

I recommend publishing the manuscript in its current form.

Author Response

This is a well-formed manuscript dedicated to the discovery and mineralogical and crystal-chemical description of the magnesian uranyl sulfate hydrate hydroxide, named chenowethite. The article is traditionally formatted and includes a detailed presentation of the morphology and optical properties of the new mineral, as well as data of electron microprobe analysis and X-ray powder diffraction. The determination of the crystal structure from a single crystal is characterized by low uncertainties, which allowed the authors to carry out a comprehensive crystal chemical analysis of chenowethite in comparison with related minerals of a similar composition.

I recommend publishing the manuscript in its current form.

Response: Thank you.